# Teleworking, Work Engagement, and Intention to Quit during the COVID-19 Pandemic: Same Storm, Different Boats?

**DOI:** 10.3390/ijerph19031267

**Published:** 2022-01-24

**Authors:** Annick Parent-Lamarche

**Affiliations:** Department of Human Resources Management, Université du Québec à Trois-Rivières, Trois-Rivières, QC G8Z 4M3, Canada; Annick.parent-lamarche@uqtr.ca

**Keywords:** teleworking, work engagement, intention to quit, individual characteristics, organizational characteristics, emotional intelligence, use of emotion, recognition, COVID-19 pandemic, labor shortage

## Abstract

The ability to retain and engage employees is now, more than ever, a major strategic issue for organizations in the context of a pandemic paired with a persistent labor shortage. To this end, teleworking is among the work organization conditions that merit consideration. The purpose of this cross-sectional study is to examine the direct and indirect effects of teleworking on work engagement and intention to quit, as well as the potential moderating effect of organizational and individual characteristics on the relationship between teleworking, work engagement, and intention to quit during the COVID-19 pandemic, based on a sample of 254 Canadian employees from 18 small and medium organizations. To address these objectives, path analyses were conducted. Overall, we found that teleworking, use of emotion, skill utilization, and recognition appear to be key considerations for organizations that wish to increase work engagement and decrease intention to quit, in the context of a pandemic paired with a labor shortage. Our results extend the literature by revealing the pathways through which teleworking, use of emotion, skill utilization, and recognition are linked to work engagement and intention to quit, and by suggesting specific interventions and formation plans that are needed.

## 1. Introduction

Accompanying the COVID-19 pandemic, a topic that has received intense media attention in the province of Quebec (Canada) is a situation of persistent labor shortages that have affected a significant number of organizations from different activity sectors [1]. These labor shortages, although they began before the pandemic, have intensified further in recent months. According to a recent study, the long-term decrease in labor force growth and the recent effects of the pandemic have worsened the impact of the labor shortages [2]. In Canada as a whole, 55% of entrepreneurs are struggling to hire the workers they need, and more than a quarter are having a hard time retaining employees [2]. In Europe, the pandemic has aggravated labor shortages in some sectors, and the problem is now emerging in others [3]. Consequently, the ability to retain and engage employees is now, more than ever, a major strategic issue for organizations. Thus, it is appropriate to develop and/or maintain employee engagement through favorable work organization conditions to face this challenge. To this end, teleworking is one of the work organization conditions that merit consideration. In fact, the COVID-19 pandemic has accelerated the modifications of work organization conditions that were already underway, such as the shift to teleworking. From April 2020 to June 2021, 30% of employees aged 15 to 64 who worked during the Labour Force Survey (LFS) reference week had worked most of their hours from home [4]. In contrast, only about 4% of employees did so in 2016 [4]. These unprecedented changes raise several questions for organizations regarding the work arrangements that should prevail once the COVID-19 pandemic is over [4]. The optimal amount of telework will likely depend on many factors, including the degree to which working from home affects worker turnover [4]; it could also depend on other work organization conditions or work characteristics that influence the effect that teleworking has on employees. Moreover, it is possible that teleworking might influence work engagement and intention to quit differently according to individual characteristics, such as personality traits and emotional intelligence, leading us to wonder if even though in the same storm (i.e., the COVID-19 pandemic), employees have been in different boats in terms of work-related and individual characteristics.

One key factor in anticipating and preventing the loss of employees is to understand the mechanisms that influence their work engagement and, consequently, determine their intention to quit. Although there exists a body of scientific literature concerning the effect of teleworking on work engagement e.g., [5,6], as well as on intention to quit e.g., [7], the former’s effect in the context of a pandemic, combined with a chronic labor shortage, is worthy of attention. Moreover, comprehending the moderating role of organizational and individual characteristics is important to better identify the optimal conditions in which to implement teleworking. Moreover, there is insufficient knowledge of the effect of new ways of working (e.g., working from home) on employees’ engagement during the COVID-19 pandemic [8]. Long before the present context arose, former U.S. President Barack Obama mentioned the importance of attracting and retaining employees who are more productive and engaged via flexible workplace policies (e.g., working from home) [9].

### 1.1. Purpose of the Study

The purpose of this cross-sectional study is to examine the direct and indirect effects of teleworking on work engagement and intention to quit, as well as the potential moderating effect of organizational and individual characteristics on the relationship between teleworking, work engagement, and intention to quit during the COVID-19 pandemic, based on a sample of 254 Canadian employees from 18 small and medium organizations (SMOs). To achieve these objectives, this study considered the Job Demands-Resources (JD-R) model [10] and the Conservation of Resources (COR) theory [11] as theoretical frameworks.

### 1.2. Background

#### 1.2.1. Work Engagement

Work engagement is defined by a positive, fulfilling, work-related state of mind marked by vigor, dedication, and absorption [12]. Vigor corresponds to high levels of energy and mental resilience while working; dedication is characterized by being intensely involved in one’s work and experiencing a sense of significance, enthusiasm, and challenge; and absorption refers to being fully concentrated and happily engrossed in one’s work. On the whole, engaged workers have high levels of energy and are enthusiastic about their job [13]. That said, in this study, we focus on the first two dimensions of work engagement defined above, namely, vigor and dedication. These two dimensions appear to be the core dimensions (the optimal measure) of work engagement [14], which is a central pillar of a worker’s well-being, which goes beyond preventing poor performance to supporting optimal functioning. The latter is essential for organizations in times of turbulence associated with the COVID-19 pandemic, combined with a context of a major labor shortage. Additionally, work engagement is seen as a core element of talent management to acquire and retain high-performing employees [15] and, as such, should be a top priority for an effective human resource (HR) system in organizations [16,17]. In light of the above, we intend to verify the effect of teleworking on work engagement, as well as its later consequences on intention to quit.

#### 1.2.2. Work Engagement and Intention to Quit

The assessment of intention to quit could be defined as an individual subjective estimation of the possibility that an individual will leave a job in the near future [18]. Intention to quit is an immediate precursor of actual turnover [19], and work engagement has been conceptualized as an antecedent of intention to quit [20,21]. Engaged workers are in a positive state of mind and immerse themselves in their work, leaving little time and space for negative thoughts, such as thinking about leaving their job [22], and previous empirical studies have found work engagement to be negatively linked to intention to quit [23,24,25].

#### 1.2.3. The Role of Teleworking

According to [26], teleworking refers to working outside the physical workplace and staying in touch with it by means of tele-communication tools or computer-based technology. This definition encompasses four types of telework: home-based telecommute, satellite offices, neighborhood work centers, and mobile work [27]. Other studies have distinguished between different forms of telework (e.g., work conducted in clients’ offices, business centers, or satellite offices). Working from home with the help of the internet and communication technologies (ICTs) is one of them [28]. For the purposes of the present study, telework corresponds to the latter situation. However, it is also important to specify that teleworking (i.e., working from home) in the context of a pandemic was, in several cases, precipitated and imposed by the health rules in force. However, even if it was implemented for contextual reasons, the trend toward teleworking is expected to continue even after the pandemic as a new way of working [29]. Indeed, it could become a work organization condition that allows for more flexibility and helps employees to achieve a better work–life balance. Furthermore, from an organizational point of view, it could be a way to attract and retain employees in a labor shortage context. 

Before the pandemic, teleworking (i.e., working from home via ICTs) presented various advantages (e.g., higher productivity, lower absenteeism, lower turnover) and disadvantages (e.g., reduced informal interaction and work coordination) for organizations [27]. Employees also experienced advantages (e.g., less time commuting, work-family balance, cost savings, reduced stress, more autonomy, and a comfortable work environment) and challenges (e.g., social and professional isolation and reduced access to resources) [27]. Furthermore, commuting, so prevalent in the pre-pandemic environment, was identified to be a work-related demand positively associated with intention to quit [7]. 

Working more hours and a reduction of the boundaries between personal and professional lives are also potential challenges associated with teleworking [30,31]. However, several empirical studies have concluded that teleworking was associated with an increase in work engagement [5,6,32]. In a recent study, researchers found that employees in the teleworking group had less stress at T2 compared to T1 [33]. Additionally, these researchers showed that teleworkers reported lower stress, lower work-to-home conflict, higher work engagement and higher job performance on teleworking days compared to non-teleworking days. To the contrary, another study found that telework was associated with a lower level of work engagement among employees [34], while yet another concluded that teleworking was not significantly associated with elevated work engagement [35]. Regarding intention to quit, ref. [36] found that teleworkers and non-teleworkers reported similar intentions to quit in a sample of American government employees. That said, a recent literature review concluded that overall, teleworking during normal times is likely to yield more positive than negative effects for employees’ health [31]. For example, telework has been associated with lower levels of emotional exhaustion [37]. 

Nonetheless, some of these findings might not apply to a pandemic context where teleworking was suddenly imposed due to exceptional circumstances. To our knowledge, no empirical studies have examined the effects of teleworking specifically on both employees’ work engagement and intention to quit during the COVID-19 pandemic. One study established that high-quality telework (e.g., agile workplaces, virtual leadership) was associated with higher work engagement during the pandemic [38]; however, different results were obtained in a study conducted by [39], which found that high intensity telework was not associated with work engagement during the pandemic. We were also able to locate a study that concluded that among employees who worked from home during the COVID-19 pandemic in Japan, an increase in sleep hours, effective interactions with supervisors, and working less than 40 hours a week were associated with high work engagement [8]. Although not specifically focusing on work engagement, one study found that most people had a more positive than negative experience of working from home during the COVID-19 pandemic [40]. The study in [41] concluded that teleworking was negatively associated with stress during the first COVID-19 lockdown, as well as positively associated with well-being [42]. Elsewhere, ref. [43] found that working from home presented the advantage of less commuting time for employees during the COVID-19 pandemic. Additionally, employees who worked from home during the pandemic appreciated the transportation time saved, the possibility to consume food and drink of their own choosing, to focus on work without interruptions, and to be close to family [44]. The difficulty of meeting with colleagues or other people, an obstacle identified by workers, was perceived as less significant in comparison to the benefits. That said, we were unable to find any study that examined the role of teleworking on intention to quit since the start of the COVID-19 pandemic. Based on the results obtained, teleworking is likely to have a positive impact on work engagement and intention to stay in one’s job (low intention to quit). Considering the important labor shortages witnessed in this pandemic, retaining and keeping employees engaged is crucial. By conducting this study, we hope to shed light on the effect of teleworking on work engagement and intention to quit.

#### 1.2.4. Effects of Individual and Organizational Characteristics

In the context of this research, the individual characteristics considered are self-esteem, locus of control, and emotional intelligence. Our choice of individual and organizational characteristics is based on previous empirical studies. These individual and organizational variables have been shown to be significantly associated with our outcomes of interest. Additionally, we were particularly interested in these variables given the fact that they could easily be targeted in a workplace and are relatively amenable to change. Self-esteem refers to an individual’s overall positive self-evaluation, which translates into an individual’s self-directed approval (higher self-esteem) or disapproval (lower self-esteem) [45]. Self-esteem was found to be positively associated with work engagement [46,47]. Additionally, self-esteem was discovered to significantly moderate the relationship between job stress and turnover intention in a previous empirical study [48]. Locus of control refers to one’s perception of one’s level of control over life events. An individual with an internal locus of control is likely to view important life events as determined by actions, efforts, or skills possessed by this individual rather than luck [49,50]. Having an internal locus of control has been found to be associated with a higher work engagement [51] and a lower intention to quit [51,52]. Having an internal locus of control has also been found to play a moderating role. More specifically, having an internal locus of control seems to moderate the association between organizational efforts and intentions to stay in the workplace [52]. Finally, emotional intelligence corresponds to a cluster of skills. These skills contribute to the accurate appraisal and expression of emotions in oneself and in others, the effective regulation of emotions, and the use of feelings to motivate, plan, and perform [53]. In addition to helping an individual to problem solve, emotional intelligence contributes to an understanding and guiding of one’s behavior [53]. Furthermore, the underlying skills that facilitate the use of emotions in adaptive ways can be learned and thereby contribute to peoples’ mental health [53]. 

Emotional intelligence encompasses four distinct dimensions: self-emotion appraisal, others’ emotion appraisal, use of emotion, and regulation of emotion. Self-emotion appraisal refers to the ability to understand one’s own emotions [54,55] and to introspect and form coherent propositions on the basis of that introspection [53]. People who accurately and quickly perceive their own emotions are capable of better expressing those emotions to others, which is essential for adequate social functioning [53]. Others’ emotion appraisal refers to the ability to perceive and understand the emotions of the people around us (also termed empathy—the ability to comprehend others’ feelings), which helps to predict others’ emotional responses [54,55] and to choose socially adaptive behaviors in response [53]. Use of emotion refers to the ability to use one’s own emotions by directing them toward constructive activities and performance (e.g., continuous self-encouragement to do better) [54,55]. Regulation of emotion refers to the ability to regulate one’s own emotions, enabling more rapid recovery from psychological distress [54,55]. This dimension is also related to the ways in which people present themselves to others to guide and control the impressions formed of them (e.g., the art of impression management), which, in turn, may lead to more adaptive and reinforcing mood states [53]. Emotional intelligence was previously found to be associated with higher levels of work engagement [56,57,58] and lower levels of intention to quit [59,60]. Another study carried out respective analyses for each dimension of emotional intelligence and established that self-emotion appraisal and use of emotion were both associated with a lower intention to quit, while others’ emotion appraisal and regulation of emotion did not significantly influence intention to quit [61]. Moreover, it was found that emotional intelligence played a moderating role in the indirect paths between perceived support from colleagues/supervisors and intention to quit [62]. That said, we were unable to locate any study that examined the moderating role of emotional intelligence on the relationship between teleworking and either work engagement or intention to quit. As previously mentioned, one key factor in anticipating and preventing the loss of personnel is understanding the mechanisms (e.g., teleworking) that influence their work engagement and, consequently, impact their intention to quit. As such, examining these relationships seems important.

In the context of this study, work characteristics encompass skill utilization, decision authority, workload, and recognition. Skill utilization refers to the possibility of using one’s skills and qualifications while having the possibility to develop new ones. Skill utilization was positively associated with work engagement [63], and a recent study confirmed that perceptions of overqualification (e.g., low levels of skill utilization) have a positive relationship with intention to quit [64]. Decision authority relates to the freedom to tackle work tasks using certain procedures at one’s own pace. Decision authority was shown to have a positive effect on work engagement, as well as a negative one on intention to quit, according to a recent empirical study on a sample of nurses [65]. Workload pertains to the quantity or difficulty of tasks, professional activities, and responsibilities. Workload was recently found to be negatively associated with work engagement [66] and positively associated with intention to quit [67,68]. Finally, recognition is a socio-emotional reward that refers to the esteem received from significant others at work (e.g., colleagues, supervisors) and appreciation received related to achievement (e.g., positive feedback) [69]. It was established that recognition was associated with higher work engagement [70,71], as well as lower intention to quit [68]

Individual and organizational characteristics have both been shown to be associated with work engagement when hindered and facilitated working from home during the COVID-19 pandemic [29]. However, we were unable to locate any study that specifically taps into the possible moderating role of work or individual characteristics (both important resources) on the relationship between teleworking, work engagement, and intention to quit since the start of the pandemic. Exploring this avenue seems relevant in order to ensure that organizations effectively orient their management practices and/or interventions. Considering that teleworking is surely here to stay, it is important to understand what can or cannot facilitate the accelerated implementation of this mode of work organization. Thus, we ask: Are there individual and/or organizational resources that may influence the effect of teleworking on employees’ attitudes and behaviors?

### 1.3. Theoretical Model

As mentioned earlier, this study considered the Job Demands-Resources (JD-R) model [10] and the Conservation of Resources (COR) theory [11] to construct its theoretical framework. One of the main overarching premises of our proposed theoretical model is that teleworking, individual resources, and organizational resources promote work engagement, which, in turn, decreases intention to quit. The JD-R model predicts outcomes through two processes: 1) demands deplete resources and can thus lead to exhaustion as well as low work engagement; and 2) resources have motivational potential and lead to better job performance (e.g., low intention to quit). Job resources (e.g., organizational characteristics) are drivers of work engagement [72]. Indeed, resourceful work environments (e.g., characterized by recognition, decision authority, and skill utilization) foster the willingness to dedicate one’s efforts and abilities to the work task [73]. A large body of research supports the association between work engagement [63,66,70,71], and employees’ intention to quit [64,65,67,68]. Individual resources (e.g., self-esteem, internal locus of control) have been found to be associated with work engagement on a theoretical [72] and empirical level [46,47,51,56,57,58]. Additionally, several empirical studies found a direct association between individual resources and intention to quit [48,51,52,59,60]. Furthermore, literature that relies on the JD-R model has established that the link between job resources and intention to quit might not be direct but instead mediated by a motivational process, such as work engagement [74]. Applied to the COR theory, organizational and personal resources could support the prevention of a cycle of loss (i.e., loss spiral), ultimately preventing employees’ from using defensive behavior (e.g., leaving or intending to quit their job). It is important to note that resources can be found within an employee as well outside this employee, such as in their work environment. To that effect, the IGLO (individual, group, leader, organizational) model offers a pragmatic classification of resources based on their provenance [75]. Resources of alternative provenance may favor work engagement and prevent intention to quit. Taken together and in accordance with the empirical background, we propose the following three hypotheses:

**Hypothesis** **1** (**H1**): *Teleworking, individual characteristics, and organizational characteristics are directly associated with work engagement*.

**Hypothesis** **2** (**H2**): *Work engagement, teleworking, individual characteristics, and organizational characteristics are directly associated with intention to quit*.

**Hypothesis** **3** (**H3**): *Work engagement mediates the relationship between 
teleworking and intention to quit*.

A second important overarching premise of our proposed theoretical model is that individual and organizational characteristics either accentuate or attenuate the effect of teleworking on work engagement, which, in turn, results in a higher or lower level of intention to quit. According to the COR theory, individuals with a greater pool of resources (e.g., favorable organizational and individual characteristics) are less vulnerable to resource loss and more capable of resource gain [76], a cycle that determines employees’ successful adaptation to their work environments. Individuals can differ in their appraisal of work organization conditions [77] (in this study, in their teleworking), leading to differences in their work engagement and intention to quit. Work characteristics can influence the effect of teleworking on work engagement and intention to quit. Additionally, work characteristics are resources that can help employees maintain a high level of functioning [75] and achieve their full potential. Resources of alternative provenance may either accentuate or attenuate the impact of teleworking on work engagement and intention to quit. In combination with teleworking, organizational and individual resources could strengthen work engagement, leading, in turn, to a lower intention of exhibiting defensive withdrawal behavior (i.e., intention to quit). Accordingly, we proposed the following three additional hypotheses: 

**Hypothesis** **4** (**H4**): *Individual characteristics (i.e., self-esteem, internal locus of control, and emotional intelligence) moderate the association between teleworking and work engagement*.

**Hypothesis** **5** (**H5**): *Organizational characteristics (i.e., decision 
authority, skill utilization, workload, and recognition) 
moderate the association between teleworking and work 
engagement*.

**Hypothesis** **6** (**H6**): *The relationship between teleworking, work 
engagement, and intention to quit is moderated by 
individual and organizational characteristics*.

See Figure 1 (hypothetical model).

## 2. Materials and Methods

### 2.1. Procedure and Participants

The data were collected during the COVID-19 pandemic in the province of Quebec, Canada (from 1 June 2020, to 5 June 2021). The sample recruited was comprised of 254 Canadian employees from 18 organizations. These organizations were recruited with the collaboration of the public affairs department of the University of Quebec in Trois-Rivières. More specifically, several organizations received a promotional e-mail specifying the objectives of the study. In addition, participating organizations received a personalized profile (“HR Profile”) of the general perception of their employees regarding different organizational aspects (e.g., work organization conditions). This profile was an important source of feedback to help them to adapt and readjust their practices (if needed) in order to retain their employees in times of a pandemic combined with a labor shortage. Everything was conducted in accordance with the strictest ethical rules for research and with the respect and agreement of all participants in the study. To this end, participants read the necessary instructions pertaining to confidentiality and signed an informed consent form prior to the completion of the questionnaire (hardcopy and online versions were available). No financial compensation was given, but participants had a chance to win a $50 gift card. The organizations included in this study were from secondary (*n* = 6) and tertiary (*n* = 12) economic activity sectors. In some, a workers’ union (*n* = 4) was present, whereas in others, there was none (*n* = 14). The average size of the organizations was 26.83 employees. For each organization, all employees were eligible to fill out a questionnaire (final response rate: 74.63%). After deleting cases with missing values, the final sample was 50.79% female, with a mean age of 41.48 years old, with 62.60% employees who were teleworking (which correspond to 159 employees) and 37.4% who were working at their usual place of work (which correspond to 95 employees).

### 2.2. Measures

#### 2.2.1. Work Engagement

The Utrecht Work Engagement Scale, shortened version (UWES-6) [78], was used to assess work engagement with a six-item seven-point additive scale with responses to each item (e.g., “At my job, I feel strong and vigorous”; α = 0.91) ranging from 0 (Never) to 6 (Daily). This corresponds to a two-dimension structure (with vigor and dedication merged together), as some previous studies confirmed that a two-dimensional scale can be chosen for academic analysis with one general score [14,79,80,81,82].

#### 2.2.2. Intention to Quit

Intention to quit was measured with a three-item seven-point additive scale with responses to each item (e.g., “I planned to look for a new job over the next 12 months”; α = 0.91) ranging from 1 (Very strongly agree) to 7 (Do not at all agree) [83].

#### 2.2.3. Teleworking

Teleworking was measured with a single item (i.e., “Which statement best describes how you perform your work during the COVID-19 crisis?”), which was coded either as 0 (“I go to my usual place of work”) or 1 = (“I work from home”).

#### 2.2.4. Individual Characteristics

The Rosenberg Self-Esteem Scale [45] was used to measure self-esteem with a six-item (e.g., “You are able to do things as well as most other people”; α = 0.81) five-point additive scale with responses ranging from 1 (Strongly agree) to 5 (Strongly disagree). For the purpose of assessing locus of control, a scale developed by [84] was used. This scale consists of a seven-item (e.g., “There is really no way you can solve some of the problems you have”; reverse coded; α = 0.79) five-point additive scale with replies ranging from 1 (Strongly agree) to 5 (Strongly disagree). Emotional intelligence was measured based on the Wong and Law EI Scale [85]. This scale comprises four distinct dimensions that were all measured on a four-item seven-point additive scale, with answers ranging from 1 (Very strongly agree) to 7 (Do not at all agree). The four dimensions assessed were self-emotion appraisal (e.g., “I have a good understanding of my own emotions”; α = 0.83), others’ emotion appraisal (e.g., “I have a good understanding of the emotions of people around me”; α = 0.89), use of emotion (e.g., “I am a self-motivating person”; α = 0.82), and regulation of emotion (e.g., “I have a good control of my own emotions”; α = 0.89).

#### 2.2.5. Organizational Characteristics

The Job Content Questionnaire [86] was used to assess decision authority with three items (e.g., “My job allows me to make a lot of decisions on my own”; α = 0.78) and skill utilization with six items (e.g., “I have the opportunity to develop my own special abilities”; α = 0.73). The Effort-Reward Imbalance Questionnaire [87] was used to measure workload with five items (e.g., “I have many interruptions and disturbances while performing my job”; α = 0.79) and *recognition* with five items as well (e.g., “I experience adequate support in difficult situations”; α = 0.84). Each variable from both questionnaires was measured on a four-point additive scale with responses ranging from 1 (*Strongly agree*) to 4 (*Strongly disagree*). 

#### 2.2.6. Control Variables

Previous studies have identified variables associated with work engagement and/or intention to quit. These variables are age [88,89,90], gender [88,90,91], marital status [88,90], and parental status [92]. Moreover, we controlled for the stress related to the COVID-19 pandemic since data were collected during the pandemic and previous studies found that stress was associated with job performance [41].

Age was coded in number of years. Gender was coded as either 0 (=male) or 1 (=female). Marital status was coded as 0 (=single) or 1 (=living as a couple). Parental status was coded as 0 (=No) or 1 (=Minor children [under 18 years of age] living with the respondent). Stress related to the COVID-19 pandemic was measured with a single item (i.e., “How has the COVID-19 crisis affected your stress level?”) and was coded as 0 (=The COVID-19 crisis decreased my stress level or did not change my stress level) or 1 (=The COVID-19 crisis increased my stress level).

### 2.3. Data Analysis

Path analyses were conducted with MPlus software (Muthén & Muthén, Los Angeles, CA, USA), [93]. These analyses allowed for the evaluation of direct and indirect associations (mediation) based on [94]. Path analysis, a subcategory of structural equation modeling (SEM), allows researchers to infer and test a sequence of causal associations between several variables and, as such, is considered an extension of multiple regression [95]. In other words, path analyses are useful for obtaining a better understanding of the processes and mechanisms underlying a given phenomenon. More specifically, the method in [94] allowed us to determine whether the association between teleworking, individual characteristics, and organizational characteristics with intention to quit were mediated by employees’ work engagement. Data analysis was carried out as follows. Teleworking, individual characteristics, organizational characteristics, as well as control variables were entered into a first model to examine their main effects on both intention to quit and work engagement. Second, teleworking, individual characteristics, and organizational characteristics were entered into a second model to test whether they had an indirect effect on intention to quit via work engagement. Third, interaction variables were entered in a final model to verify if individual and organizational characteristics played a moderating role in the relationship between teleworking and work engagement. A two-tailed probability for rejection of the null hypothesis (*p* ≤ 0.05) was considered in order to determine the significance levels of the combined variables, as well as for each individual regression coefficient. Models were tested with maximum likelihood estimation using robust standard errors (MLR estimation). The goodness-of-fit was assessed using the comparative fit index (CFI) and the Tucker–Lewis index (TLI). Values above 0.95 for the CFI and TLI indicated an excellent fit [96].

## 3. Results

### 3.1. Descriptive Analyses

Table 1 provides descriptive statistics (mean/proportion, standard deviation) and correlations for the main variables of the study. 

### 3.2. Multiple Regression Analyses

Table 2 provides results on the main effects of work engagement, teleworking, individual characteristics, and organizational characteristics on intention to quit and the main effects of teleworking, individual characteristics, and organizational characteristics on work engagement. The results demonstrate that work engagement and recognition were associated with lower levels of intention to quit. Moreover, teleworking was associated with lower levels of work engagement, while one dimension of emotional intelligence (i.e., use of emotion), skill utilization, and recognition were associated with higher levels of work engagement. 

See Figure 2 (Final Model Results).

### 3.3. Multiple Mediation Analyses

In addition, the results presented in Table 3 include variables that indirectly influenced intention to quit via work engagement: teleworking (higher levels of intention to quit); one dimension of emotional intelligence (i.e., use of emotion), skill utilization, and recognition (lower levels of intention to quit). 

### 3.4. Multiple Moderation Analyses

No interaction emerged as significant in the final model, which was designed to verify the moderating effect of individual and organizational characteristics on the relationship between teleworking and work engagement.

## 4. Discussion

The purpose of this cross-sectional study was to examine the direct and indirect effects of teleworking on work engagement and intention to quit, as well as the potential moderating effect of organizational and individual characteristics on the relationship between teleworking, work engagement, and intention to quit during the COVID-19 pandemic, based on a sample of 254 Canadian employees from 18 SMOs.

The first hypothesis (H1), which postulated that teleworking, individual characteristics, and organizational characteristics were directly associated with work engagement, was partially supported. We found that emotional intelligence (i.e., use of emotion), skill utilization, and recognition were positively associated with work engagement, while teleworking was negatively associated with work engagement. On the other hand, emotional intelligence (i.e., self-emotion appraisal, others’ emotion appraisal, and regulation of emotion), self-esteem, locus of control, decision authority, and workload were not significantly associated with work engagement. The results are consistent with previous empirical studies on emotional intelligence [56,57,58], skill utilization [63], and recognition [70,71]. These results are also consistent with the JD-R model, which postulates that individual and organizational resources are drivers of work engagement [72]. That said, the results pertaining to the effect of teleworking were surprising and contrary to what was expected. Considering that it was previously established that most people had a more positive than negative experience of working from home during the COVID-19 pandemic [40,44] and that teleworking was negatively associated with stress during the first COVID-19 lockdown as well as positively associated with well-being [41,42], we expected teleworking to be associated with higher work engagement. Moreover, most studies prior to the pandemic had found that teleworking was associated with work engagement [5,6,32,33,35]. It is possible to argue that this result might be partially explained by the sample of this study. Indeed, our sample was comprised of employees of SMOs. Such organizations are generally characterized by relational and hierarchical proximity (i.e., all employees being directly linked to the manager), as well as functional proximity (i.e., having fuzzy borders between functions) [97]. Thus, the relations are highly personalized and informal, and there is a relative absence of formalized HRM tools [97]. Hence, it is possible that these organizations did not have formal procedures for teleworking, for instance. Additionally, employees of SMOs might be accustomed to frequent interactions, proximity management, spatial proximity [98], as well as informal communications [99], which were most probably lacking during the suddenly imposed teleworking of the pandemic (as it was less well organized since it was unplanned). Additionally, it should be noted that the data for this study were collected after the first wave of the pandemic and during the second and third waves in the province of Quebec, Canada (from 1 June 2020, to 5 June 2021). Therefore, it is possible that some employees felt that they had had enough of the public health measures, including mandatory teleworking. Over that period of time, it is likely that the pandemic and the restrictive public health measures had drained the employees, especially since they had fewer opportunities for socialization for a prolonged period of time. There was still no prospect of a clearly defined end to the pandemic at the time of the data collection, even though vaccination started halfway through this period. Actually, 55.12% of the employees in our sample indicated that the COVID-19 pandemic had increased their stress level (which was controlled for in our analysis). All things considered, the effect of teleworking in this study was different from the rather positive effect during the first wave/lockdown and before the pandemic.

The second hypothesis (H2) postulated that work engagement, teleworking, individual characteristics, and organizational characteristics were directly associated with intention to quit. This hypothesis was also partially supported. More precisely, work engagement and recognition were both negatively associated with intention to quit. Teleworking, emotional intelligence, self-esteem, locus of control, decision authority, skill utilization, and workload were not significantly associated with intention to quit, whereas the results pertaining to work engagement [23,24,25] and recognition [68] are consistent with previous empirical studies. Moreover, these results are in line with the JD-R model, which postulates that resources have a motivational potential and lead to improved job performance (e.g., low intention to quit). That said, we expected teleworking to be negatively associated with intention to quit, but our result did not reach statistical significance. As commuting was identified as a work-related demand positively associated with intention to quit [7], and considering that teleworkers did not have to commute, we believed that teleworking would be a resource (in the same way that commuting was identified as a demand) negatively associated with intention to quit. 

The third hypothesis (H3), which postulated that work engagement mediated the relationship between teleworking and intention to quit, was partially supported. We observed that teleworking, emotional intelligence (i.e., use of emotion), skill utilization, and recognition were indirectly associated with intention to quit via their effects on work engagement. More precisely, emotional intelligence (i.e., use of emotion), skill utilization, and recognition were associated with a lower level of intention to quit via their effect on work engagement. Teleworking was associated with a higher level of intention to quit because of its effect on work engagement. These results indicate that the effects of teleworking, emotional intelligence (i.e., use of emotion), skill utilization, and recognition on work engagement were still strong enough to influence intention to quit indirectly. Even though we were unable to locate any study that examined the mediating role of work engagement on the relationship between teleworking, individual characteristics, and organizational characteristics with intention to quit, these results are not surprising. Indeed, literature that relies on the JD-R model has established that the link between job resources and intention to quit might not be direct but instead mediated by a motivational process, such as work engagement [74]. Moreover, and according to the COR theory, organizational and personal resources could enable the prevention of a cycle of loss (i.e., a loss spiral), ultimately preventing employees from using defensive behavior (e.g., leaving or intending to quit the job). The only surprising result is the one pertaining to the indirect effect of teleworking on intention to quit. An association was found between teleworking and higher intention to quit via the former’s effect on work engagement. Only one study had previously found teleworking to be directly associated with lower job performance during the first pandemic lockdown [41]. Again, this result may be explained by our sample and data collection timing, as mentioned above (see H2). Moreover, it is important to keep in mind that the pandemic context carries several specificities regarding teleworking [100]. These specificities may explain the surprising indirect association between teleworking and intention to quit identified in this study. Among those specificities, the shift to teleworking was sudden, involuntary, and not anticipated by employees and employers [100]. As a result, employees may not have had the equipment and resources needed to maintain their work engagement and, therefore, their intention to stay (e.g., ergonomic workstations, high-performance computers, closed offices, printers, and cameras for videoconferencing). Additionally, telework occurred in a stressful context (i.e., the COVID-19 pandemic) paired with increased social isolation due to social distancing orders [100].

The fourth hypothesis (H4), which postulated that individual characteristics (i.e., self-esteem, internal locus of control, emotional intelligence) moderated the association between teleworking and work engagement, was not supported. We found that individual characteristics did not significantly moderate the association between teleworking and work engagement. This is not in line with previous studies that found that self-esteem [48], locus of control [52], and emotional intelligence [62] played a moderating role at work. Moreover, this result is not in accordance with the COR theory, which postulates that individuals with a greater pool of resources (e.g., favorable individual characteristics) are less vulnerable to resource loss and more capable of resource gain.

The fifth hypothesis (H5) postulated that organizational characteristics (i.e., decision authority, skill utilization, workload, and recognition) would moderate the association between teleworking and work engagement. This hypothesis was not supported. We observed that none of the organizational characteristics had a significant moderating effect on the relationship between teleworking and work engagement. It was expected that organizational characteristics would (by the same reasoning as for the fourth hypothesis), to the contrary, play a moderating role. Indeed, it was established that individual and organizational characteristics both had a positive relationship with work engagement when working from home during the COVID-19 pandemic [29].

The sixth hypothesis (H6) postulated that the relationship between teleworking, work engagement, and intention to quit would be moderated by individual and organizational characteristics. This hypothesis was also not supported since no moderating effects were identified. Therefore, no moderated mediation effects were found. No study, to our knowledge, has investigated the moderating effects of individual and organizational characteristics on the relationship between teleworking and work engagement, nor has any study verified whether this entails repercussions for intention to quit. We anticipated that those characteristics would generate a significant additional gain of resources, leading to higher work engagement and lower intention to quit, corresponding with COR theory [11].

Overall, we found that work engagement played a mediating role between teleworking, individual characteristics, and organizational characteristics with intention to quit. Teleworking was directly associated with a lower level of work engagement and indirectly associated with a higher intention to quit. Additionally, we found that use of emotion (a dimension of emotional intelligence) was directly associated with higher work engagement and indirectly associated with lower intention to quit. Additionally, skill utilization and recognition were both directly associated with higher work engagement, as well as indirectly associated with lower intention to quit. Recognition was also associated with lower intention to quit. Therefore, teleworking, use of emotion, skill utilization, and recognition appear to be important considerations for organizations that wish to increase work engagement and decrease intention to quit in a context of a pandemic paired with one of labor shortage. 

### 4.1. Practical Implications

Our findings suggest that practitioners should pay special attention to teleworking, use of emotion, skill utilization, and recognition. These were found to be determinants of work engagement and intention to quit in this study. First, teleworking practices and policies should be rethought. To ensure that teleworking, again, becomes a positive factor for employee work engagement both during the pandemic and afterward, employers should reach out to employees daily, if only to maintain social contact [101]. Moreover, employers should ensure continuous communication about expectations, work progress, and availability [102]. They should also offer employees the flexibility to organize their work schedules and priorities and provide them with good technological equipment in order to facilitate their engagement. It is possible that poor implementation of teleworking conditions can become discouraging over time and demotivating for employees. Therefore, better-informed implementation and official work-at-home policies are suggested. Accordingly, high-quality telework characterized by agile workplaces and virtual leadership was associated with higher work engagement during the pandemic [38]. For instance, we suggest surveying employees about their preferences regarding the number of teleworking days per week, work schedules, equipment and resources needed, and ways to maintain social contact while teleworking. Second, use of emotion should also be targeted by practitioners. This emotional intelligence dimension is associated with adaptive problems solving. The study in [53] referred to four dimensions of the ability to solve problems: flexible panning (e.g., the dominant affect of emotionally intelligent people is positive, and positivity is more likely to facilitate the generation of a large number of plans for themselves), creative thinking (e.g., positive mood facilitates creativity), redirected attention (e.g., attention is directed to new problems when powerful emotions occur that help to reprioritize and allocate attentional resources accordingly), and motivation (e.g., positive moods motivate persistence at challenging tasks as they increase confidence in one’s ability to face obstacles and aversive experiences). Fortunately, these can all be easily developed with the help of training and interventions [103]. Accordingly, such interventions should be implemented via a training plan to ensure the development of these important competencies among employees. Third, skill utilization should also be the target of organizational interventions aimed at ensuring that employees’ strengths are challenged by their supervisors [104]. Fourth, recognition is vital, and organizations should make sure that programs are not just “another box for managers to check” [105]. Instead, appreciation needs to be specific and genuine, with leaders who take the time to engage with employees authentically [105]. Additionally, official recognition programs should include highlighting employees’ successes, organizing an employee recognition day, recognizing years of service with gifts, etc. 

### 4.2. Limitations and Recommendations for Future Research

This study, as with all studies, has some limitations. First, the possibility of common variance bias must be acknowledged, since all variables were collected from the same source. Despite this possibility, we believe that the risk of common variance bias was low, given the diversification of our sample in terms of different organizations (*n* = 18). Indeed, the organizations included in our sample operated in both secondary and tertiary economic sectors, and only some had unionized employees. Second, data were collected with self-report questionnaires completed by employees who volunteered to participate in the study. Complementing the self-reported data with interviews with supervisors could have limited response bias. Moreover, interviewing supervisors could have allowed us to converge their perceptions with those of employees with respect to the work organization conditions, for instance. Third, one also needs to acknowledge the possibility of selection bias due to the organizations that decided to participate. These organizations are decidedly considerate of their employees: the simple fact of voluntarily participating in such a study to receive a personalized “HR Profile” for their employees in order to better orient their practices in the future was a sign of goodwill and commitment on their part. Fourth, in light of the above, our results may not necessarily be generalizable to the general workforce. Fifth, even though we controlled for gender in our statistical analysis, the same variables could influence men and women differently. These patterns of associations with gender should be investigated in future studies. Additionally, future studies should attempt to further investigate the effects of teleworking in these times of accelerating and perpetual changes. Particular attention should be paid to the alignment between the person/employee and teleworking. Perhaps teleworking is not for everyone. In this regard, we believe that emotional intelligence or other specific skills should be examined in relation not only to adaptation to new working methods, but also to whether individuals thrive using new methods. It would also be possible to examine teleworking from the perspective of the self-determination theory [106] and the basic psychological needs for autonomy, competence, and relatedness. For instance, we might wonder if different levels or preferences in terms of these basic psychological needs influence the effect of teleworking on work engagement and intention to quit (e.g., lower need for relatedness might influence differently the effect of teleworking on work engagement and intention to quit). Future studies could control for the period of administration of the questionnaire and look for differences between participants who completed the study at different time points (e.g., different waves of the pandemic). Additionally, it could be interesting to verify whether the quality of work life of employees, who continued to work at their usual place of employment during the pandemic, differed from those who started working from home. Other factors that could play a role in work engagement and intention to quit should be examined in the future. Amongst these factors are human resource management practices, being the head of a household (i.e., needing a pay check) and home responsibilities that morphed during the pandemic (i.e., young children at home, elderly care, etc.). Finally, it would be interesting to replicate the present study with other samples in terms of organizations, employees, and countries.

## 5. Conclusions

The main objectives of this study were to examine the direct and indirect effects of teleworking on work engagement and intention to quit, as well as the potential moderating effect of organizational and individual characteristics on the relationship between teleworking, work engagement, and intention to quit during the COVID-19 pandemic. The results obtained indicate that even in the same storm, it appears that employees were in different boats in terms of teleworking, use of emotion, skill utilization, and recognition. Although we are aware of the inherent limitations of this study, we still hope to contribute to the many current reflections on the subject. Our results extend the literature by showing the pathways through which teleworking, use of emotion, skill utilization, and recognition are linked to work engagement and intention to quit, and by suggesting specific interventions and/or formation plans that are needed. Undeniably, there is a risk that there will be other pandemics and a multiplication of variants of the COVID-19 virus in the future (at the time of writing, the OMICRON variant is a good example of this) that will, in turn, plunge organizations and their employees into imposed teleworking or prolong its use. One takeaway is that although this study shows that teleworking seems to harm the engagement and retention of employees, it will nevertheless be necessary to ensure that teleworking is favorable to their health, motivation, engagement, and intention to stay in the future. The context of the pandemic combined with that of the labor shortage is not about to change and it is imperative to adapt to it the best way possible. Among other considerations, perhaps the optimal implementation of telework and work organization conditions, as well as training for developing competencies such as emotional intelligence, should take precedence.

## Figures and Tables

**Figure 1 ijerph-19-01267-f001:**
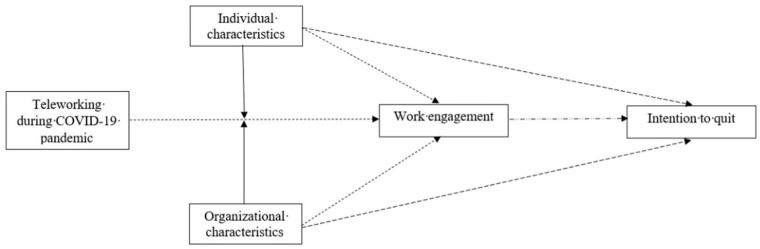
Hypothetical Model.

**Figure 2 ijerph-19-01267-f002:**
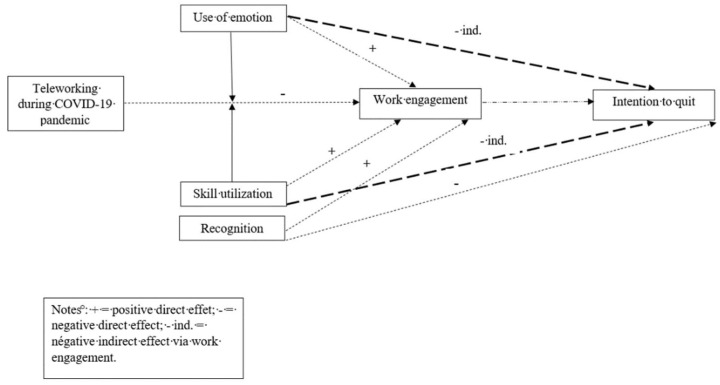
Final Model Results.

**Table 1 ijerph-19-01267-t001:** Descriptive correlational statistics.

	M	SD	1.	2.	3.	4.	5.	6.	7.	8.	9.	10.	11.	13.	14.
1.	6.81	5.00	1												
2.	25.56	7.23	−0.50 **	1											
3.	62.60		0.07	−0.06	1										
4.	22.92	3.42	0.01	0.24 **	0.08	1									
5.	21.70	4.52	0.01	0.18 **	0.03	0.38 **	1								
6.	22.87	3.89	−0.09	0.37 **	−0.03	0.44 **	0.14 *	1							
7.	22.29	4.33	−0.06	0.22 **	0.04	0.26 **	0.08	0.38 **	1						
8.	25.59	3.03	−0.10	0.33 **	0.10	0.39 **	0.19 **	0.53 **	0.40 **	1					
9.	28.39	4.19	−0.24 **	0.32 **	0.14 *	0.24 **	0.20 **	0.39 **	0.36 **	0.50 **	1				
10.	9.29	1.75	−0.22 **	0.21 **	0.17 **	0.08	0.14 *	0.22 **	0.20 **	0.32 **	0.37 **	1			
11.	17.95	2.91	−0.25 **	0.27 **	0.28 **	0.03	0.14 *	0.10	0.11	0.18 **	0.26 **	0.51 **	1		
12.	11.85	3.04	0.08	−0.08	0.11	−0.05	0.09	−0.08	−0.10	−0.00	0.08	0.04	0.18 **	1	
13.	16.66	2.64	−0.30 **	0.28 **	0.18 **	0.16 **	0.16 **	0.12	0.29 **	0.25 **	0.37 **	0.44 **	0.33 **	−0.19 **	1

Note a: * *p* ≤ 0.05 (coefficients ≥ 0.05) and ** *p* ≤ 0.01 (coefficients ≥ 0.05). Note b: M = Mean/Proportion; SD = Standard deviation; 1. = Intention to quit; 2. = Work Engagement; 3. = Teleworking; 4. = Self-emotion Appraisal; 5. = Others’ Emotion Appraisal; 6. = Use of Emotion; 7. = Regulation of Emotion; 8. = Self-esteem; 9. = Locus of control (internal); 10. = Decision authority; 11. = Skill utilization; 12. = Workload; 13. = Recognition.

**Table 2 ijerph-19-01267-t002:** Direct Effects of Teleworking, Individual and Organizational Variables on Work Engagement and Intention to Quit.

	Work Engagement	Intention to Quit
Constant	16.515 **	18.661 **
WORK ENGAGEMENT		
Work Engagement		−0.312 **
TELEWORKING		
Teleworking	−2.398 **	0.957
INDIVIDUAL VARIABLES		
Self-Emotion Appraisal	0.071	0.103
Others’ Emotion Appraisal	0.068	0.103
Use of Emotion	0.438 **	0.089
Regulation of Emotion	0.003	0.061
Self-esteem	0.250	0.149
Locus of control (internal)	0.105	−0.154
ORGANIZATIONAL VARIABLES		
Decision Authority	−0.474	−0.124
Skill Utilization	0.678 **	−0.128
Workload	−0.075	0.008
Recognition	0.481 *	−0.353 **
ADJUSTMENTS		
CFI	1.00 1.00 173.598 (33) **	
TLI		
χ^2^ (df)		

Note. * *p* ≤ 0.05 and ** *p* ≤ 0.01. The following variables were controlled for: age, gender, marital status, parental status, stress related to the COVID-19 pandemic (unstandardized coefficients).

**Table 3 ijerph-19-01267-t003:** Indirect Effects of Teleworking, Individual, and Organizational Variables on Intention to Quit.

	Estimate	SE
TELEWORKING		
Teleworking	0.747 *	0.299
INDIVIDUAL VARIABLES		
Self-Emotion Appraisal	−0.022	0.046
Others’ Emotion Appraisal	−0.021	0.028
Use of Emotion	−0.137 *	0.055
Regulation of Emotion	−0.001	0.039
Self-esteem	−0.078	0.059
Locus of control (internal)	−0.033	0.035
ORGANIZATIONAL VARIABLES		
Decision Authority	0.148	0.084
Skill Utilization	−0.211 **	0.068
Workload	0.023	0.044
Recognition	−0.150 *	0.072

Note. * *p* ≤ 0.05 and ** *p* ≤ 0.01. The following variables were controlled for: age, gender, marital status, parental status, stress related to the COVID-19 pandemic (unstandardized coefficients).

## Data Availability

The data are not publicly available in order to respect the privacy of research participants.

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
