# Peer review of "Teleworking, Work Engagement, and Intention to Quit during the COVID-19 Pandemic: Same Storm, Different Boats?"

_ijerph, 2022, doi:10.3390/ijerph19031267_

Round 1

Reviewer 1 Report

Dear Author,

I read this interesting work that placed attention to employees work engagement during work from home and its relationship with intention to quit, by also considering the moderating role of organizational and personal demands and resources.

A thorough review of the literature has been conducted by the author to support the hypothesis. However I would highloght some concerns with thoeretical expostion and development of hypothesis. Specifically, the theoretical model concerning the JDR and COR theory should be placed before the literature review in order to guide the hypothesis developement conserning the relationship of the variable and the model tested in the present study. Furthermore authors should better define differenced between job and personal resources and how they are conceputalized within the theoretical models and their relationship with both outcomes and job demands.

Another aspect concerns the way by which the authors support their hypothesis or justify the relevance of investigating such relationships: in addition to underlining the fact that there are no other studies that have focused on these relationships (eg. p. 4 line 154,  p. 5 line 210 etc.) the authors should be more convincing about the relevance of analyzing these relationships. Furthermore, in my point of view section 

1.2.3 and 1.2.4 result to be excessively dispersive to the readers. Therefore, I suggest to summarize more the results highlighted by the literature and to connect these evidences to the development of hypotheses that should be placed after exposing the empirical evidence regarding each construct.

Rgarding participant, authors should report how many workers were on teleworking and how many of them were going to the usual palce of work during the covid pandemic.

Methods are clearly exposed and statiscal analyses are well developed.

Regarding this, I suggest to control for the period of administration of the questionnaire. Indeed, if the data were collected between june 2020 and june 2021, significant differences may have occurred, in terms of well-being and adaptation to the pandemic, between those who completed the questionnaire in the first period, compared to those who completed the questionnaire in the middle of the second and third waves.

there is a typo at page 11 line 465 (

we expected teleworking to be negatively associated with teleworking, but our result did not reach statistical significance.

Authors should also report a figure with the final model, namely depicting the significant paths emerged from the analyses.

Finally, I think thas some reflections should be made on the differences between who were on teleworking and who do not. This should underline another relevant topic, that is to say, the theme concerning the quality of working life of those who continued to work during the pandemic and that had to face other challenges confronted to those who started to work from home.

Reviewer 2 Report

This cross-sectional study investigated the effects of teleworking on work engagement and intention to quit, as well as the potential moderating effect of organizational and 12 individual characteristics on the relationship between teleworking, work engagement, and intention to quit during the COVID-19 pandemic. Given the current labor landscape, this paper is timely and the examines telework from various angles. This reviewer finds the methodology to be appropriate, although some variables that may significantly affect teleworking were not measured, such as 1) head of household (i.e. need for job) and home responsibilities that shifted with the pandemic (i.e. young children at home, elderly care, etc.). A closer proof-read is recommended, as there are some typos (lines 106, 107, 180). 

Reviewer 3 Report

This is a very interesting and well-written paper. It contributes to the field of teleworking with a useful empirical study. There are some minor issues that deserve some further consideration.

It is certainly important to discuss contextual factors such as the Covid-19 pandemic, but I am wondering if it is necessary to consider the labor shortage issue, especially at the beginning of the study. How does it affect teleworking, work engagement, or the other key variables of the study?    

l84: You mean the first two dimensions - not "to dimensions"

You define work engagement with all three dimensions but then focus only on two of them.  The justification for this could be brought forward, the first time you explain why you do so. 

The section on teleworking is very well informed. Since findings are quite mixed, it would be helpful to include a brief comment with your overall assessment of the impact of teleworking on chosen outcomes.

Section 1.2.4 begins with one very long paragraph. Consider breaking it down. In this paragraph, emotional intelligence seems to be defined twice.

l.180 - check the word positively

The rationale for including the specific individual and organizational variables could be a little clearer from the beginning of the section. You use the JDR model and state that all of these characteristics are resources, yet I am wondering why you focus on the specific resources. Why should each of the six variables be considered as a moderator? Is the justification the same for all variables?

In terms of sampling, if you focused on 18 organizations with an average size of 26.83, then the population is 486.  Your sample is 254 employees.  Hence, why is the final response rate 74%? It should be lower.

Although this is not my field of expertise, I am surprised to see that gender was only presented with two answers. Shouldn't there be non-binary options included here?

l.465 - "we expected teleworking to be negatively associated with teleworking" needs to be rephrased.

l519-521 - how was it established that individual and organizational characteristics 'both hinder and facilitate working from home?" Is the absence of moderation sufficient to claim this?

The discussion is thoughtful and concise.  Practical implications and limitations are well explained. 
